# Early Sowing on Some Soybean Genotypes under Organic Farming Conditions

**DOI:** 10.3390/plants12122295

**Published:** 2023-06-12

**Authors:** Victor Petcu, Ancuța Bărbieru, Mihaela Popa, Cătălin Lazăr, Laurențiu Ciornei, Amalia Gianina Străteanu, Ioana Claudia Todirică

**Affiliations:** 1Centre of Studies and Research of Agroforestry Biodiversity, Academy House, Romanian Academy, 050711 Bucharest, Romania; laurentiu.ciornei@cscbas.ro (L.C.); amalia.strateanu@cscbas.ro (A.G.S.); ioana.todirica@cscbas.ro (I.C.T.); 2National Agricultural Research and Development Institute Fundulea, Călăraşi County, 915200 Fundulea, Romania; ancuta.barbieru@incda-fundulea.ro (A.B.); mihaela.popa@ricic.ro (M.P.)

**Keywords:** organic, soybean, early sowing, Romania, plant density, cumulative stress index, weeds

## Abstract

The demand for soybeans in Europe motivates breeders, researchers, and growers to find suitable cultivars to adapt and extend the soybean crop to improper climate areas. Weed control is a crucial aspect of crop technology in organic agriculture, but particularly for soybean crops. In laboratory conditions, the cumulative stress index for seedlings was determined to identify the susceptible cultivars. A field experiment with 14 soybean accessions and 2 sowing dates was conducted under organic farming conditions over the course of three years, from 2020 to 2022. Plant population density was found to be significantly (*p* < 0.01 and *p* < 0.1) negatively correlated to the degree of resistance to low temperature as well as infestation degree with weeds (for *p* < 0.05 and *p* < 0.1), with the exception of early sowing in 2021. Yield was significantly (*p* < 0.05, *p* < 0.01, *p* < 0.1) correlated with plant population density, with the exception of optimal sowing in 2022. Early sowing variants emerged with vigor in the first two years, breeding lines and registered varieties showed low input, and organic agriculture systems showed low yields in the drought years of 2020 and 2022. Although early sowing even in the first two years proved to be a practice that increased the cultivars’ performance, in 2022, due to the long period of chilling stress in the field, this option had negative effects on yield due to the high weed frequency. Therefore, the early sowing strategy for the soybean crop in this particular case of non-irrigated conditions in a temperate continental area proved to be a risky practice.

## 1. Introduction

The importance of worldwide soybean (Glycine max L. Merrill) cultivation has risen with the increasing demand for protein sources for livestock. The grain’s lysine amino-acid content of over 5% makes this plant an important protein crop [1] and, as an oil plant, it stands besides sunflower and palm oil [2]. Although other pulses have notable quality traits [3], soybean remains the widest cultivated grain legume plant, both worldwide and at the European level [4]. Additionally, the use of soybean plants extends to its use as a high-quality forage crop [5,6], vegetables (known as ‘edamame’ (Japanese)), ‘mao dou’ (Chinese), ‘Poot kong’ (Korean), beer beans, sweet beans, and green soybeans (in other parts of the world) [7]. Epoxidized soybean oil is a raw material for biodegradable polymers [8], and it is notable that in human consumption, soybean peptides have positive effects on chronic diseases such as obesity, diabetes, and cardiovascular problems [9]. Furthermore, soybean cultivation improves soil quality and local biodiversity [10,11], but on the global level, extending the soy crop by deforestation is causing environmental and social problems [12].

Thus, agronomists are being driven by the rising demand for soybean products in Europe to broaden cultivation restrictions and investigate new crop production conditions [13,14].

However, the concern with the future and potential of expanding soybean-cultivated areas is on local climate conditions; the farm profitability of the cattle, poultry, and pig sectors, which are the main protein users [15,16,17]; and the post-harvest processing and food industry development within a country [18].

In Romania, Bulgaria, and the Republic of Moldova, countries with a recent history of about 100 years of soybean cultivation have a potential of at least one million hectares for soybean crops [19].

The cultivation of soybeans in organic farming or agroecological systems is linked with premium quality and non-GMO [20] products for human consumption.

Organic farming needs suitable cultivars and technologies to perform quality and yield demands, and even more so as the control of weeds, specific pathogens, and pests is different from conventional agriculture, where there are possibilities for chemical suppression [21,22,23]. Weed management is a key factor in crop technology. Soybeans are sensitive in the early stages [24], in which the weeds are more adapted to lower temperatures.

Optimal planting dates for soybeans vary according to variety, cropping system, and environmental conditions, but the delay in typical sowing date is one of the organic techniques for effective weed control in the soybean crop [25]. On the other hand, the early sowing of soybeans has attracted the attention of researchers and farmers because it offers benefits in capitalizing on early precipitation, avoids drought and high temperatures during mid-summer (when plants are in a critical stage of development), prevents the attack of insects at the end of the growing season, and the crop can be harvested earlier (shake losses are avoided) [26,27,28].

Early sowing, as a technological alternative for soybean cultivation in organic farming conditions from Romania, is little or insufficiently studied. 

In Central and South European countries, such as Romania and Bulgaria, the major risk to soybean crops is the high water demand of crops; therefore, maintaining the proper moisture in the upper soil layer of 0–40 cm is strongly recommended for gaining yield and quality [29,30]. The critical period for water is from flowering to pod physiological maturity [31], which, in many areas of Romania, coincides with periods of drought and heat. 

At early sowing, farmers may risk yield loss from the poor establishment of the crop due to low soil temperature. However, due to the continuous increase in global air temperature over the years, there is a tendency for soybeans to be sown earlier.

It is very important to determine the cold tolerance of soybean varieties in the early stages of growth. Today, many different soybean varieties are available in the world in terms of tolerance to low temperatures. In Romania, there are limited data on the suitability of soybean varieties for early sowing, and further local research is needed in order to help the breeding process for regional adaptation of the cultivars [32].

Therefore, the choice of varieties, together with other technological factors (such as the time of planting), are essential for profitable and healthy [33] soybean crops in Romania under the country’s particular environmental conditions.

The vegetation phase, usually expressed by local conditions in the maturity group and stem type of growth (either determinate or indeterminate, for the chosen cultivars in the study), reveal different responses in the field, depending on the genotype x environment complex interactions [34,35].

The purpose of this study was to highlight the effect of early sowing on some soybean genotypes, sown under the conditions of the organic farming system.

## 2. Results

### 2.1. Climate Conditions

The experimentation years varied in terms of total rainfall, ranging from 180 to 269.2 mm in the period from April to August, as well as in terms of monthly repartitions of these measurements.

In 2020 and 2022, the moisture deficits from June up to August created unfavorable conditions during the appearance of the reproductive organs and grain formation.

In 2021, the rainfall during June exceeded by 60.1 mm the normal level for the zone (74.9 mm), suggesting favorable conditions for the soybean crop; however, in July and August a moisture deficit of 49.9 mm and 25.3 mm vs. the, multi-annual average was registered, the time when soybeans are in the flowering (R1–R2) to full-seed (R6) stage and, thus, most vulnerable to water stress (Table 1).

The 2020 and 2021 growing seasons were the warmest (0.5 °C and 1.8 °C above normal). In 2021, the month of April was the coolest (1.6 °C below normal), but the 2020 growing season was cool, especially during the first decade of May when the average minimum temperature was 5.5 °C below the same period in 2021 and 2020 (Table 1).

### 2.2. Effect of Sowing Time, Genotype on Soybean Yields

The analyses of the variance highlighted the very significant effect of sowing date and genotype on soybean production, and in dry years, the interaction between the two factors influenced the obtained production (Table 2).

Early sowing had a positive impact on the soybean production during the first two years of experimentation (Table 2). The yields achieved in 2021, which was normal in terms of the rainfall that was recorded, were on average 222 kg ha^−1^ higher than those attained when sowing during the ideal season, whereas the early sowing in 2022 reduced the production on average by 500 kg ha^−1^ (Table 3).

#### Yield Loss Due to Weeds

The results of the analysis of variance showed that the weed infestation was very significantly affected by the weather conditions of the experimental years, soybean genotypes, and also by interactions between these factors both for early sowing and optimal sowing (Table 4).

The degree of weed infestation was between 19 and 30% for optimal sowing and between 25 and 35% for early sowing. There were significant negative correlations between yields and weed distribution on plots (r = −0.62 **, r = −0.73 ***) (Figure 1).

Yields in 2020 and 2022 were low due to the drought which caused accelerated, irregular ripping as well as a high frequency of weeds. The productions obtained from early sowing in 2022, a very dry year with low temperatures recorded in spring, were only 48% of the production obtained from sowing at the optimal time. The weeds’ distribution per plot this year was very high (59–85%). Moreover, the weeds’ occurrence was correlated with the yields obtained (Figure 2 and Figure 3).

There is a complex weed spectrum in all crop development stages with monocot weeds such as *Setaria viridis*, *Echinochloa crus-galli*, and *Digitaria sanguinalis* and dicot weeds such as *Convolvulus arvensis, Fallopia convolvulus, Amaranthus retroflexus, Portulaca oleracea, Solanum nigrum, Ambrosia artemisiifolia,* and *Chenopodium album* (Table 5).

In the last year, the estimated weed density was very high, which means competition for nutrients and water from the soil. This is even more so in the maturity phase at early sowing, when the weed density was 228 weeds m^−2^ and the largest share was *Ambrosia artemisiifolia* and *Amaranthus retroflexus* (Table 4). The studies conducted by Patterson and Flint [36], demonstrated that *Amaranthus* sp., with C4 metabolism, showed higher WUE compared to soybean plants. Due to global warming, the abundance of common ragweed (*Ambrosia artemisiifolia* L.) began to increase in Southern, Central France, and Northern Italy as well as in Romania [37,38]. In the organic farming system and not only there, it will be very difficult to combat weeds, especially because in addition to the large seeded production, there is a high long-term survival of seeds.

### 2.3. Plants Density and Cumulative Stress Index

The results of the analysis of variance showed that the weed infestation was very significantly affected by the weather conditions of the experimental years, soybean genotypes, as well as by interactions between these factors both for early sowing and optimal sowing (Table 6).

The density of the plant population decreased with later sowing and in dry years. It is clear that the density of the plant population was generally lower in the soybean varieties that are more vulnerable to low temperatures (cumulative stress index with higher values) than in the resistant ones (Table 7).

Low temperature is one of the primary abiotic stresses, which negatively affects the growth and productivity of soybean. The identification of soybean genotypes with tolerance to low temperature is important for the genetic improvement of soybean stress tolerance as well as for the choice of genotypes suitable for early sowing [39].

Plant population density was found to be significantly (*p* < 0.01 and *p* < 0.1) negatively correlated to the degree of resistance to low temperature as well as with estimated weed density (for *p* < 0.05 and *p* < 0.1), with the exception of early sowing in 2021.

Yield was significantly (*p* < 0.05, *p* < 0.01, *p* < 0.1) correlated with plant population density, with the exception of optimal sowing in 2022 (Table 8).

These correlations provide valuable information for breeding new soybean, respectively, for the consolidation of favorable traits affecting the technological and utilitarian value of plants, such as yield potential or resistance to cold stress.

## 3. Discussion

The negative effect of water restriction in soybean plants depends on the phenological stage [40]. Drought in the spring affects sowing, which often results in uneven emergence, lack of seedlings, and poor seedling growth [41].

In our study, the experimental years were very different regarding rainfall. So, if in 2020 there was insufficient rainfall, (180 mm during the entire growing period and only 14 mm in April), which primarily affected the emergence and the density of the plants, instead in 2021, the total rainfall registered was approximately 269 mm (47.6 mm in April), ensuring sufficient soil moisture for a uniform emergence of the soybean genotypes studied. This explains the differences in production obtained in the years of experimentation, both at early sowing and at the optimal time. Moreover, data from the specialized literature show that unlike cereals, legume plants need more water at the beginning of the vegetation period to germinate and emerge. 

Water availability is usually higher on early seeding conditions, but the temperature requirements of soybean plants are above those of weed species.

Therefore, varieties resistant to low temperatures are needed to be able to be sown early. Thus, work is being completed to improve the resistance of genotypes to low temperatures in the soybean improvement programs from different parts of the world (including Romania). The yield differences, as an interaction between genotype variations and specific environmental conditions is a common research topic for soybean breeders [42,43,44]. 

The temperatures immediately after sowing and in the first stage of the vegetation are very important for early sowing. One of the main risks of planting very early is that the emerged plants will be damaged by cold temperatures, as soybeans are sensitive in the cotyledon stage. In this way, the risk that comes from the fact that the germination and appearance of the cotyledons are delayed at lower soil temperatures is mitigated [45].

We used genotypes with different resistance to low temperatures in this study. In genotypes with a lower cumulative stress index for low temperatures, it was found that the emerged plants were much less damaged by cold temperatures, so that plant population density was higher compared with the sensitive ones. However, changes in temperature occurred in 2022 (the temperature for the first decade of May 2022 was 5.5 °C) at the two chosen sowing times, resulting in significantly different plant numbers and overall grain yield, which led to excessive growth of weeds. As is known, the degree of weed infestation is also affected by meteorological factors, such as moisture and temperature [46]. Our data demonstrate a significant relationship between sowing date and the estimated weeds’ number. Other studies show that the intensity of weed competition may vary according to the density and composition of weed species present in the agricultural area, as well as the competitive ability of the variety used, soil and crop management practices, and the period of coexistence between the crop and weed community [47,48]. Thus, the interference of weeds in the crop can cause reductions of up to 80% in grain yield. In our case, the weed interface in the early sown soybean crop simultaneously with the cold and drought (in 2022) caused reductions in soybean production of up to 60%. So, although early sowing even in the first two years has proven to be a practice that increased the cultivars’ performance, in 2022, due to the long period of chilling stress in the field, this option had negative effects on yield due to the weeds’ infestation. Therefore, the early sowing strategy for the soybean crop in non-irrigated conditions is a risky practice in organic farming in continental temperate areas.

Optimization of sowing dates is the most important and least expensive agronomic practice that affects soybean yield. Some researchers have suggested that earlier sowing dates have contributed to recent soybean yield gains in the United States. For example, sowing soybeans in late April and early May is currently recommended in the Midwestern United States to achieve the maximum seed yield [49].

In order to be able to sow soybeans earlier, efforts are being made to obtain genotypes resistant to low temperatures during seed germination and the first phases of vegetation. The genotypes studied by us showed genetic variability for resistance to low temperatures; five of the genotypes were sensitive according to the cumulative stress index. However, the level of resistance is not high enough to compensate for the negative effects resulting from early sowing in the organic farming system.

Early sowing even in the first two years proved to be a practice that increased the cultivars’ performance; however, in 2022, due to the long period of chilling stress in the field, this option had negative effects on yield due to the infestation with weeds. Therefore, the early sowing strategy for the soybean crop is a risky practice in organic farming in continental temperate areas.

Further research and field studies for expanding the soybean crop in Europe’s continental areas could be made by inter-cropping or finding the best technologies for planting soybeans as a secondary crop. Romania is an important maize producer [50], a maize–soybean intercropping system is a solution that could enhance the land equivalent ratio and improve the resilience of the entire agroecosystem by mitigating the risk of a total crop monoculture failure [51,52].

## 4. Materials and Methods

Field experiments were conducted over three growing seasons during 2020, 2021, and 2022 at the National Agricultural Research and Development Institute (44°26′ N; 26°30′ E), on cambic chernozem soil type.

Two management factors, very important for organic soybean farming, were investigated: (1) soybean varieties and (2) sowing time.

The experience was bifactorial of type 2 × 14 in 3 repetitions, with A factor (sowing times: a1—sown early; a2—sown optimally) and B (genotypes: b1 … b14). This experience was carried out under organic farming conditions, on cambic chernozem, well drained, formed on loess, with 33.8% clay content and 2.8% organic matter in arable layer.

Two sowing times were used, first an early sowing at the beginning of April (2–3) and second an optimal sowing time, two weeks after the first sowing.

In the study, 14 soybean genotypes were used: 4 registered varieties and 10 breeding lines, from maturity group 0 to 00. Eleven of them possessed a determined growing stem, one of them with an indeterminate stem growing and two with semi-determinate growing type (Table 9).

The plot dimension was 9 m^2^. Distance between rows was 50 cm, with seeding rate of 55 germinable seeds m^−2^. The seeds were treated with a product accepted in organic farming, 10% CuSO_4_ solution. The applied tillage system (ploughing in autumn and a three disc harrow in the spring in each year) was uniform and no other inputs such as fertilizers, biostimulants, or bacterial treatments that could influence the crop production [53,54].

Two mechanical weeding works, and two manual weeding sessions were performed in the vegetation period. No irrigation or other plant protection products had been applied on the plots. Therefore, the applied technology was organic low-input.

### Measurements

Soybean yield, expressed at 10% humidity, was determined by eliminating protective areas and harvesting the entire plot with a plot combine harvester. After that, a grain analyzer was used to determine the water content. 

Weed density and cover were measured by counting the number of weeds per plot inside a frame of 0.25 m^2^ in dynamics at three stages of plant development, in three replicates for each plot. Degree of infestation with weeds (%) for each genotype was visually estimated in all plots (replicates), both in the early sowing variant and control variant (sowing in optimal time).

Soybeans were sampled three times during the growing season: early season at the beginning of weed competition (trifoliate leaves), mid-season at peak crop growth (full flowering—R2), and at the beginning of maturity (R7) [55] for determining the weeds species.

At full flowering and pod setting, the percent cover of the crop and each weed species were estimated visually within each frame.

The density (plants m^−2^) was determined by counting the number of plants from a 0.25 m^2^ frame and multiplied by 4. 

Names and abbreviations of weed species could be found on the EPPO database [56].

In the laboratory, a cold germination test was used to evaluate the seeds’ vigor and the ability of seeds to produce normal seedlings under cold conditions. The method consisted of using a soil paste (soil moistened with 60% of its water-holding capacity) applied on a wet thick paper towel. Then, the seeds (100 in four replicates) are counted and placed on the towel and covered with a wet thin paper towel. The rolled paper towels were placed in a chamber with cold temperature of 6 °C for 7 days. The seedlings were transferred to a chamber at a temperature of 25 °C for an additional 4 days. Germ assessment was carried out after the completion of the 11 days based on the international norms regarding seed quality testing (ISTA-2006) and the ISTA Germ Assessment Manual.

In parallel, the warm germination test (control, at 25 °C) was carried out.

The germinative faculty and vigor elements, hypocotyl length, radicle length, and germ weight, were analyzed.

The length of hypocotyl and radicle were measured on each seed directly using a ruler. Dry weight of germ was determined after drying at 105 °C overnight.

The cumulative stress index (CSI) for low temperatures was calculated as the sum of the relative individual component responses at cold and optimal temperatures, according to the formula described by Koti et al. [57]
CSI = [(HLc − HLo)/HLo + (RLc − RLo)/RLo + (GWc − GWo)/GWo + (GFc − GF0)/GFo × 100]
where HL represents the hypocotyl length; RL is the radicle length; GW represents the germ weight; and GF is the germinative faculty at cold (c) and optimal (o) temperatures.

Statistical analysis of the data was performed by analysis of variance [58] calculated in Excel and by correlation analysis. The correlation coefficients (r) were calculated based on the linear regression analysis through the Excel program.

## Figures and Tables

**Figure 1 plants-12-02295-f001:**
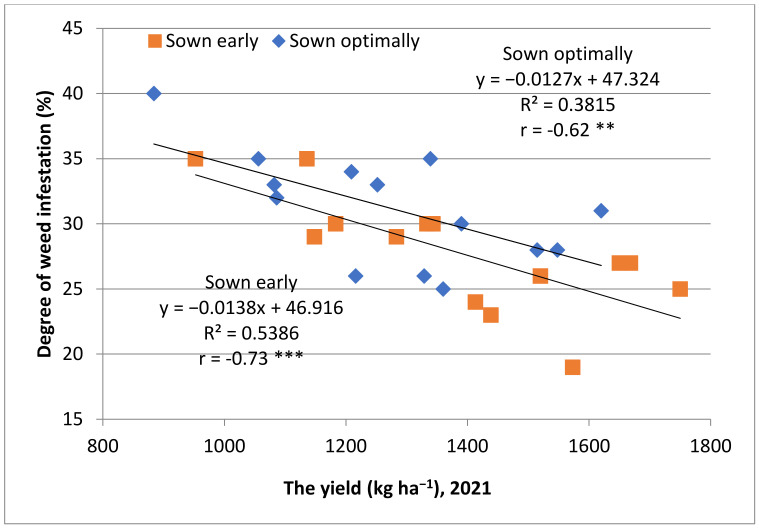
Relationships between yields and degree of weed infestation, 2021; ** moderate negative correlation; *** high negative correlation.

**Figure 2 plants-12-02295-f002:**
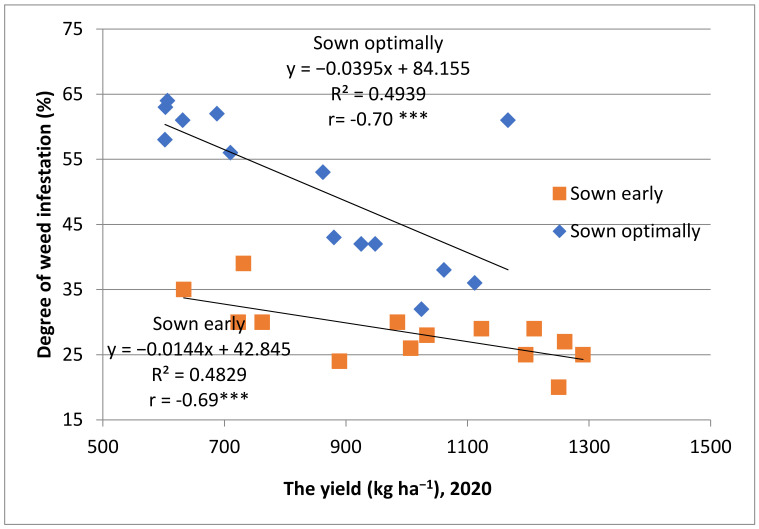
Relationships between yields and degree of weed infestations, 2020; *** high negative correlation.

**Figure 3 plants-12-02295-f003:**
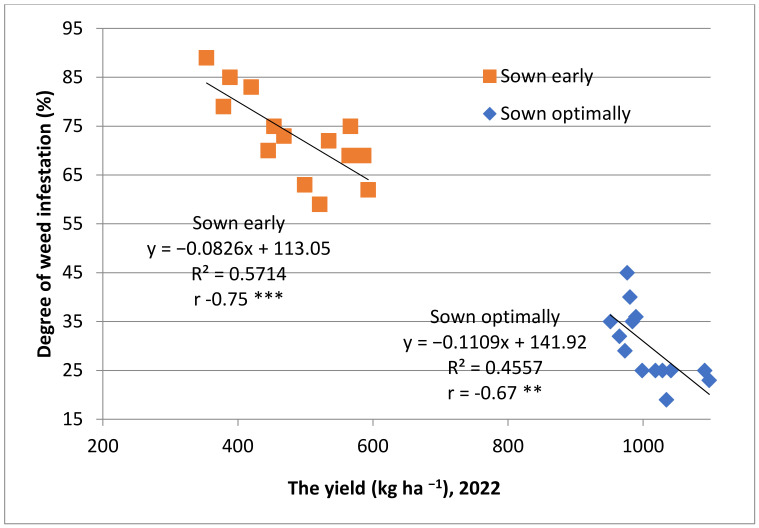
Relationships between yields and degree of weed frequency, 2022; ** moderate negative correlation; *** high negative correlation.

**Table 1 plants-12-02295-t001:** Monthly distribution of rainfall and mean air temperature during the soybean growing season in 2020, 2021, and 2022 including the 60-year average at the study site.

Month	Year	Temperature (°C)	Rainfall (mm)
April	60-years average	11.3	45.1
	2020	12.3	14
	2021	9.7	31
	2022	12.1	47.6
May *	60-years average	17	62.5
	2020	17	58
	2021	17.2	57.6
	2022	17.9	30.1
June	60-years average	20.8	74.9
	2020	21.7	68.4
	2021	21.1	135
	2022	22.6	59.6
July	60-years average	22.7	71.1
	2020	25.1	34.2
	2021	25.3	21.2
	2022	25.0	29.2
August	60-years average	22.3	49.7
	2020	25.5	5.4
	2021	24.2	24.4
	2022	25.6	14.4
Total/Mean	60-years average	18.8	303.3
	2020	20.3	180.0
	2021	19.5	269.2
	2022	20.6	180.9

* The temperatures for the first decade of May: 2020 (10 °C), 2021 (10.12 °C), 2022 (5.5 °C).

**Table 2 plants-12-02295-t002:** Analyses of Variance for the effect of sowing time and genotypes on soybean yields.

Source of Variance	Factor F and Significance		
	2020	2021	2022
Sowing time (Factor A)	313.08 ***	70.30 ***	72.00 **
Genotypes (Factor B)	50.14 ***	52.52 ***	12.06 ***
Interaction A × B	2.32	1.32	3.23 ***

** significant as *p* < 0.05; *** highly significant as *p* < 0.001.

**Table 3 plants-12-02295-t003:** Yield results of 14 soybean genotypes on optimal and early sowing in 2020, 2021, and 2022.

Genotype	Year	Yield (kg ha^−1^) Sown Early	Yield (kg ha^−1^) Sown Optimally	Yield Difference %	% Yield Difference (kg ha^−1^)
F10-1443	2020	732	602	21.6	130
	2021	1148	1082	6.1	66
	2022	453	952	−52.4	−499
F13-993	2020	1136	1056	7.6	80
	2021	723	606	19.3	117
	2022	445	977	−54.5	−532
F13-1174	2020	1413	1339	5.5	74
	2021	889	710	25.2	179
	2022	388	984	−60.6	−596
F14-878	2020	1183	1086	8.9	97
	2021	762	688	10.8	74
	2022	354	981	−63.9	−627
	2020	953	884	7.8	69
F14-918	2021	633	603	5	30
	2022	379	989	−61.7	−610
F13-1114	2020	1520	1390	9.4	130
	2021	1007	1025	−1.8	−18
	2022	567	1041	−45.5	−474
F13-1124	2020	1439	1329	8.3	110
	2021	873	632	38.1	241
	2022	420	1035	−59.4	−615
F13-908	2020	1573	1515	3.8	58
	2021	1033	948	9	85
	2022	535	1018	−47.4	−483
F15-749	2020	1333	1252	6.5	81
	2021	885	862	2.7	23
	2022	468	973	−51.9	−505
F15-792	2020	1750	1620	8	130
	2021	1157	1167	−0.9	−10
	2022	565	1029	−45.1	−464
Anduța F	2020	1650	1548	6.6	102
	2021	1088	1061	2.5	27
	2022	499	965	−48.3	−466
Flavia	2020	1283	1209	6.1	74
	2021	867	880	−1.5	−13
	2022	593	1091	−45.6	−498
Larisa TD	2020	1343	1216	10.4	127
	2021	919	1024	−10.3	−105
	2022	586	1098	−46.6	−512
Teo TD	2020	1668	1360	22.6	308
	2021	1196	1112	7.6	84
	2022	521	844	−38.3	−323

**Table 4 plants-12-02295-t004:** Analyses of Variance for the effect of year and genotypes on weed infestation.

Source of Variance	Factor F and Significance	
	Sown Early	Sown Optimally
Year of experimentation (Factor A)	227.98 ***	112.46 ***
Genotypes (Factor B)	25.88 ***	99.83 ***
Interaction A × B	19.87 ***	29.15 ***

*** highly significant as *p* < 0.001.

**Table 5 plants-12-02295-t005:** Effect of year and sowing time on the weed infestation (plants m^−2^) in soybean crops (mean of all genotypes).

Phase of Vegetation	Variants	Year	SETVIR	ECHCG	DIGSA	CONAR	POLCO	AMARE	POROL	SOLNI	AMBEL	CHEAL	TOTAL
Trifoliate leaves	Early	2020	96			3		36		25			160
		2021	125				14			63			202
		2022	150			21	14	96		179			460
	Optim	2020	68				7	14					89
		2021	96			41	28	35		7			207
		2022	115			50		40		10			215
Full flowering and pod setting (R2)	Early	2020	7								10	8	25
		2021	28	12	10	13		7			15	10	95
		2022	30	15	15	14		10			20	15	119
	Optim	2020	14								10		24
		2021	30	10	8	7					15	7	77
		2022	35	18	11	8					20	15	107
Maturity start (R7)	Early	2020	35	7	7	7					10		66
		2021	30	30	25			6	6	2	15	14	128
		2022	45	40	35	29		15	11	15	25	13	228
	Optim	2020	7								10		17
		2021	25	26	18	22		9		7	15		122
		2022	55	41	35	30	15	12		9	9		206

SETVIR—*Setaria viridis*; ECHCG—*Echinochloa crus-galli*; DIGSA—*Digitaria sanguinalis*; CONAR—*Convolvulus arvensis*; POLCO—*Fallopia convolvulus*; AMARE—*Amaranthus retroflexus*; POROL—*Portulaca oleracea*; SOLNI—*Solanum nigrum*; AMBEL—*Ambrosia artemisiifolia*; CHEAL—*Chenopodium album*.

**Table 6 plants-12-02295-t006:** Analyses of Variance for the effect of year and genotypes on plants’ density.

Source of Variance	Factor F and Significance	
	Sown Early	Sown Optimally
Year of experimentation (Factor A)	18.77 ***	29.13 ***
Genotypes (Factor B)	56.21 ***	69.58 ***
Interaction A × B	11.43 ***	13.64 ***

*** highly significant as *p* < 0.001.

**Table 7 plants-12-02295-t007:** Plant population density and correlation coefficients among studied traits for soybean genotypes.

Genotype	Years	Early Sowing (Plants m^−2^)	Optimal Sowing (Plants m^−2^)	Cumulative Stress Index
	2020	27	25	
F10-1443	2021	37	35	0.83
	2022	21	25	
	2020	25	32	
F13-993	2021	31	29	0.91
	2022	20	24	
	2020	37	33	
F13-1174	2021	35	37	0.71
	2022	26	33	
	2020	37	31	
F14-878	2021	34	36	0.83
	2022	24	29	
	2020	25	24	
F14-918	2021	41	40	0.86
	2022	23	26	
	2020	32	39	
F13-1114	2021	45	40	0.44
	2022	32	33	
	2020	40	41	
F13-1124	2021	45	45	0.47
	2022	26	36	
	2020	51	47	
F13-908	2021	55	49	0.3
	2022	28	30	
	2020	40	42	
F15-749	2021	42	42	0.47
	2022	34	36	
	2020	43	44	
F15-792	2021	39	52	0.26
	2022	31	32	
	2020	33	36	
Anduța F	2021	52	50	0.45
	2022	34	34	
	2020	34	36	
Flavia	2021	52	52	0.43
	2022	32	28	
	2020	47	39	
Larisa TD	2021	50	53	0.42
	2022	36	39	
	2020	45	43	
Teo TD	2021	50	46	0.31
	2022	33	36	

**Table 8 plants-12-02295-t008:** Correlation coefficients among studied traits for soybean genotypes.

Correlation Coefficient	Years	Early Sowing	Optimal Sowing
Crop density (plants m^−2^) × Cumulative Stress Index (CSI)	2020	−0.76 ***	−0.89 ***
	2021	−0.75 ***	−0.84 ***
	2022	−0.81 ***	−0.67 **
Crop density (plants m^−2^) × Weeds density (plants m^−2^)	2020	−0.51 *	−0.48 *
	2021	−0.44	−0.56 *
	2022	−0.56 *	−0.68 **
Cumulative Stress Index (CSI) × Weeds density (plants m^−2^)	2020	0.57 *	0.64 **
	2021	0.65 **	0.64 **
	2022	0.54 *	0.86 ***
Yield × Crop density (plants m^−2^)	2020	0.65 **	0.71 ***
	2021	0.48 *	0.50 *
	2022	0.71 ***	0.36

* low correlation; ** moderate correlation; *** high correlation.

**Table 9 plants-12-02295-t009:** Soybean varieties and breeding lines used in the field experiment.

N.	Soybean Genotype	Maturity Group	Stem Type	Maintainer
1	F10-1443	0	Det.	NARDI Fundulea
2	F13-908	00	Det.	NARDI Fundulea
3	F13-993	00	Det.	NARDI Fundulea
4	F13-1114	00	Det.	NARDI Fundulea
5	F13-1124	0	Det.	NARDI Fundulea
6	F13-1174	00	Det.	NARDI Fundulea
7	F14-878	00	Det.	NARDI Fundulea
8	F14-918	0	Det.	NARDI Fundulea
9	F15-749	0	Det.	NARDI Fundulea
10	F15-792	0	Det.	NARDI Fundulea
11	Anduța F	0	Det.	NARDI Fundulea
12	Florina F	0	InDet.	NARDI Fundulea
13	Larisa TD	0	SemiDet.	SCDA Turda
14	Teo TD	00	SemiDet.	SCDA Turda

Det. = Determined growing; InDet = Indeterminate growing; SemiDent = Semi-determinate growing; NARDI Fundulea = National Agricultural Research and Development Fundulea; SCDA Turda = Station of Agricultural Research Turda.

## Data Availability

The data presented in this study are available on request from the corresponding author.

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
