# Peer review of "Early Sowing on Some Soybean Genotypes under Organic Farming Conditions"

_plants, 2023, doi:10.3390/plants12122295_

Round 1

Reviewer 1 Report

The article has represented a very important study which can help in adapting soybean in nonconventional areas. I have many concern about the study as provided below:

"Therefore, the early sowing strategy for soybean crops, in non-irrigated conditions it is a risky practice in organic farming in continental temperate areas" - here this is confusing, why do authors want to specify the organic farming conditions?

While the study highlights the effects of genotypes and early sowing on soybean crops under organic farming conditions, it is important to note that there were no controls to evaluate the effect of conventional (non-organic) farming practices. Therefore, to draw conclusive evidence on the effects of early sowing, a replicated evaluation should be conducted under both organic and conventional farming conditions. This would help to determine if the early sowing strategy is indeed a risky practice specifically under organic farming conditions or if it applies equally to conventional farming conditions

Figure 1, 2, and 3: figure legends are missing, what are the orange and blue spots? 

Also better to combine these three figures and a multi-panel figure 

"Experimental design was in RCB (Randomized Complete Blocks), in two replicates" - Only two replicates are not sufficient for such studies. 

no comment 

Author Response

Dear reviewer,

Thank you very much for your comments.

As much as could, we have improved the paper. 

We have added the legends of the figures, it will be to crowded to bind them.

Please find it attached.

Kind regards,

Victor Petcu

Reviewer 2 Report

Dear Editor and Authors,
Thank you very much for the request to make this review. The review is in attachment.

Kind regards

Reviewer

Author Response

Dear reviewer,

Thank you very much for your comments.

As much as could, we have improved the paper. 

Please find the modification marked in green.

Kind regards, 

Victor Petcu

Reviewer 3 Report

1. „known as edamame” (L: 37) – vegetable soybean has many local names. See „Vegetable soybeans are also known as ‘edamame’ (Japanese), ‘mao dou’ (Chinese), ‘Poot kong’ (Korean), beer beans, sweet beans, and green soybeans (in other parts of the world) - https://doi.org/10.3390/plants12030609. Here the authors gave only one of these local names.

2. Introduction - please indicate clearly the research hypothesis and the aim of the research.

3. L: 93-94. It should be specified that the given rainfall sums (180 to 269.2 mm) refer to the period from April to August.

4. Standardise the notation: rainfall (e.g. L: 93) or precipitation (L 119).

5. The notation of units of measurement should be in accordance with the SI system.

6. I suggest giving the seed yield in tons per hectare.

7. Table 7 – please insert below the table an explanation of the abbreviations used in the table

8. Were the seeds inoculated with Bradyrhizobium japonicum bacteria before sowing to enhance the effect of BNF?

9. “Weed density and cover were measured by counting the weeds number per plot inside a frame of 0.25 m2” (L: 278-279) – the weed evaluation was carried out for each variant of the experiment (1 repetition) or on each plot (2 repetitions) ?

10. "Weeding intensity (%) was visually estimated." (L: 279) - in this case, what was the control?

11. L: 306 – please identify the author (literature citation) of this formula.

12. In Discussion, the authors state the facts and compare them with the results of other authors, but there is no attempt to explain the occurring phenomena.

13. Item 16 of the literature deals with the effect of the yeast Rhodotorula rubra upon egg yolk colour and layer hens' productive performances, not with soybean nutrition.

14. In my opinion, item 20. “Chivu, L.; Ciutacu, C. About Industrial Structures Decomposition and Recomposition. Econ proceedings. Finance 2014, 375 8, 157–166,” is not thematically related to the presented research topic. In addition, the data contained therein cover the years 1990-2011.

Author Response

Dear reviewer,

Thank you very much for your comments.

As much as could, we have improved the paper. 

Please find the modifications marked in red.

Kind regards,

Victor Petcu

Round 2

Reviewer 1 Report

No further comment

Author Response

Thank you very much for the review. We have made just a few minor changes to satisfy all reviewers.

Reviewer 2 Report

Accept in present form

Author Response

(The authors gave the same response as above.)

Reviewer 3 Report

- Oznaczenie jednostek miary powinno być zgodne z układem SI, np. Ryc. 1 - 3 „kg/ha”, Tablica 5 „liczba/m2”, Tablica 7 „rośliny/m2”, nasiona / m2 "

- L 509 „densitywas” – błąd ortograficzny

Author Response

Thank you very much for your comments and suggestions we have fixed the orthography and notation into SI suggested mode.
